# Humoral Immune Response Following COVID-19 Vaccination in Multifocal Motor Neuropathy and Chronic Inflammatory Demyelinating Polyneuropathy

**DOI:** 10.3390/vaccines13090902

**Published:** 2025-08-26

**Authors:** Louise Sloth Kodal, Sonja Holm-Yildiz, Sebastian Rask Hamm, Laura Pérez-Alós, Cecilie Bo Hansen, Mia Marie Pries-Heje, Line Dam Heftdal, Rasmus Bo Hasselbalch, Johannes Roth Madsen, Ruth Frikke-Schmidt, Linda Maria Hilsted, Erik Sørensen, Sisse Rye Ostrowski, Henning Bundgaard, Peter Garred, Kasper Iversen, Susanne Dam Nielsen, John Vissing, Tina Dysgaard

**Affiliations:** 1Copenhagen Neuromuscular Center, Department of Neurology, Rigshospitalet, University of Copenhagen, 2100 Copenhagen, Denmarktina.dysgaard@regionh.dk (T.D.); 2Viro-Immunology Research Unit, Department of Infectious Diseases, Rigshospitalet, Copenhagen University Hospital, 2100 Copenhagen, Denmarksusanne.dam.poulsen@regionh.dk (S.D.N.); 3Laboratory of Molecular Medicine, Department of Clinical Immunology, Section 7631, Rigshospitalet, Copenhagen University Hospital, 2100 Copenhagen, Denmark; 4Department of Cardiology, Rigshospitalet, Copenhagen University Hospital, 2100 Copenhagen, Denmark; 5Department of Cardiology, Herlev and Gentofte Hospital, Copenhagen University Hospital, 2100 Copenhagen, Denmark; 6Department of Clinical Biochemistry, Rigshospitalet, Copenhagen University Hospital, 2100 Copenhagen, Denmark; 7Department of Clinical Medicine, Faculty of Health and Medical Sciences, University of Copenhagen, 2100 Copenhagen, Denmark; 8Department of Clinical Immunology, Rigshospitalet, Copenhagen University Hospital, 2100 Copenhagen, Denmark; 9Department of Emergency Medicine, Herlev and Gentofte Hospital, Copenhagen University Hospital, 2100 Copenhagen, Denmark

**Keywords:** immune response, COVID-19, vaccine, Chronic Inflammatory Demyelinating Polyneuropathy, CIDP, Multifocal Motor Neuropathy

## Abstract

Background/Objectives: Multifocal Motor Neuropathy (MMN) and Chronic Inflammatory Demyelinating Polyneuropathy (CIDP) are immune-mediated polyneuropathies often treated with immunoglobulin therapy. They were prioritized for COVID-19 vaccination during the pandemic. However, their immune response following COVID-19 vaccination remains unclear. We investigated short- and long-term immune responses to COVID-19 vaccination in patients with MMN and CIDP compared to controls. Methods: In a prospective observational study, patients with CIDP or MMN and matched controls were followed over 24 months. Controls were age- and sex-matched 1:9. Participants received COVID-19 vaccines in accordance with the Danish vaccination program. Primary outcomes were levels of SARS-CoV-2 IgG antibodies and virus-neutralizing capacity. A positive vaccine response was defined as IgG > 225 AU/mL and neutralizing capacity ≥ 25%. Results: We included 34 patients and 306 matched controls. While baseline SARS-CoV-2 IgG levels were similar, controls exhibited higher IgG levels at 6- (mean difference, 88%; *p* = 0.008), 18- (91%; *p* = 0.023), and 24 months (160%; *p* < 0.001). Neutralization capacity was also higher in controls at 6 (10%, *p* = 0.004), 18 (7%, *p* < 0.001), and 24 months (9%, *p* = 0.002). Despite this, the proportion of vaccine responders did not differ between the two groups after 24 months (*p* = 0.196). In patients receiving immunoglobulin therapy, IgG levels were lower than in controls at 24-month follow-up alone (56%, *p* < 0.001); all demonstrated a positive vaccine response. Conclusions: Patients with CIDP and MMN demonstrated a positive humoral response to COVID-19 vaccination. Although IgG and neutralization levels were lower than in controls, all patients receiving immunoglobulin therapy were vaccine responders.

## 1. Introduction

Multifocal Motor Neuropathy (MMN) and Chronic Inflammatory Demyelinating Polyneuropathy (CIDP) are both immune-mediated polyneuropathies. Their exact pathogenesis is not fully understood. CIDP and MMN both involve different immunopathological mechanisms, including cellular, humoral, and complement-mediated pathways [1]. A total of 30–50% of patients with MMN have IgM anti-GM1 autoantibodies [2]. The two polyneuropathies differ in their clinical presentation. While MMN is a motor neuropathy without sensory involvement, CIDP may involve both motor and sensory nerves [3,4]. If left untreated, the disorders may progress, leading to permanent physical impairments due to reduced muscle strength and loss of sensation.

For CIDP, first-line treatment options include immunoglobulin or corticosteroids. Corticosteroids are generally cheaper but associated with more long-term side effects, and treatment recommendations can vary by country [3]. In Denmark, immunoglobulin is preferred [5]. In contrast, patients with MMN do not respond to corticosteroids, and immunoglobulin is the only first-line treatment [4]. Corticosteroids affect both humoral and cellular immunity, whereas B-cell-depleting therapy impairs the humoral immune response. Immunoglobulin is an immunomodulatory therapy used to regulate the immune response in patients with immunodeficiencies or autoimmune diseases. Immunoglobulins are antibodies that function through several mechanisms, including actions on Fc receptors, and may also suppress the function of regulatory T cells [6,7]. The treatment effect of intravenous (IVIg) or subcutaneous (SCIG) can be considered equivalent.

During the COVID-19 pandemic, it was reported that severe acute respiratory syndrome coronavirus-2 (SARS-CoV-2) infection could trigger or worsen polyneuropathy symptoms in patients with MMN and CIDP [8,9]. A small cross-sectional study of 29 patients with CIDP revealed that 21% had experienced worsening of polyneuropathy symptoms after SARS-CoV-2 infection [10]. Studies such as these generated concerns among patients and physicians, and patients with CIDP and MMN were characterized as a vulnerable group, similar to individuals undergoing chemotherapy treatment [11]. Therefore, like in patients with cancer and autoimmune diseases, vaccination was offered to patients with MMN and CIDP to prevent relapse of polyneuropathy symptoms associated with SARS-CoV-2 infection.

With the rapid implementation of the COVID-19 vaccination program, concerns arose among patients and physicians regarding the potential reduction in COVID-19 vaccine response in patients receiving immune-modulating therapies compared to healthy individuals [12]. This concern was supported by studies conducted alongside the implementation of vaccination programs, which showed that immune responses to COVID-19 vaccination were impaired in patients with various autoimmune conditions who received immunosuppressive therapies compared to the general population [13,14]. One study showed that glucocorticoids and B-cell-depleting agents impair the immune response to COVID-19 vaccines in patients with autoimmune diseases [13], while another study found a delayed humoral immune response to COVID-19 vaccination in patients with myasthenia gravis compared to healthy individuals [15].

Based on studies showing a diminished response to COVID-19 vaccinations for patients receiving immunosuppressive therapy compared to the general population, an intensified vaccination protocol was recommended for individuals with MMN and CIDP. As part of this strategy, booster vaccines were offered to these patient groups. The implementation of enhanced vaccine frequency in patients with CIDP and MMN was based on a theoretical rationale due to their use of immunosuppressive therapies. However, these recommendations were made without sufficient data to support this notion, and no disease-specific or treatment-specific evidence supported this approach. The immune response following COVID-19 vaccination in patients with MMN and CIDP who are receiving immunoglobulin treatment remains unknown. These patients are considered at risk of reduced vaccine efficacy and are offered booster vaccines. On the other hand, vaccination can, in rare cases, induce worsening of CIDP and MMN symptoms. Therefore, it is important to investigate whether this population demonstrates a sufficient humoral immune response.

To this end, this study aimed to investigate short- and long-term immune responses to COVID-19 vaccination in patients with MMN and CIDP who received immunoglobulin compared to immunocompetent individuals.

## 2. Materials and Methods

### 2.1. Study Design and Inclusion Criteria

This prospective observational study was initiated at the Copenhagen Neuromuscular Center, Rigshospitalet, Copenhagen, in February 2021, inviting patients diagnosed with CIDP or MMN. It was conducted over a period of two years (2021–2023). COVID-19 vaccines were given according to the Danish vaccination program. Participation in this study did not affect the national vaccination strategy for any participant. The control group consisted of healthcare professionals from Copenhagen University Hospital, Rigshospitalet, and Herlev-Gentofte Hospital, who were also invited to participate in this study, as reported in previous studies [16]. The controls, who comprised immunocompetent individuals, were age- and sex-matched 1:9. Written and oral informed consent was obtained from all participants. Participants were monitored starting from a baseline sample obtained before or within 13 days of receiving the first COVID-19 vaccine dose and followed with subsequent blood samples collected at approximately 3 weeks, and 2, 6, 12, 18, and 24 months following the initial vaccination. Information on clinical symptoms and medication was collected from medical records.

### 2.2. Outcomes and Definitions

Information on administered vaccines and the number of vaccine doses was collected from the Danish Vaccination Register, a national register in Denmark, where all vaccinations are registered [17].

Levels of SARS-CoV-2 Immunoglobulin G (IgG) antibodies and virus-neutralizing capacity were the primary endpoints. We used the same methodology as employed in previous studies [18,19]. The IgG antibodies were specific for the receptor-binding domain (RBD) of the ancestral spike protein and were determined using an in-house enzyme-linked immunosorbent assay (ELISA). SARS-CoV-2 anti-RBD IgG levels were reported in arbitrary units (AU) per milliliter (ml). A value > 1 AU/mL was considered detectable. Values below 1 AU/mL were assigned a value of 1 AU/mL. The cut-off value for a positive IgG response was 225 AU/mL [18].

The antibody-neutralizing capacity was assessed using an in-house ELISA-based assay, which measured the degree of inhibition of the interaction between the angiotensin-converting enzyme 2 (ACE-2) host receptor and the RBD using a methodology as previously employed [19]. A cut-off value for neutralizing response was defined as 25% inhibition [20]. Our study began before the international standard was available. Therefore, we prioritized consistency within our study by using the same standards over the two years, as well as the same recombinant antibodies as calibrators to validate this study. Moreover, a positive and a negative control were included in each ELISA plate, providing robustness to the assay. Previous reports have indicated that the unavailability of international standards did not eliminate the observed differences across studies [21,22].

A positive vaccine response following COVID-19 vaccination was defined as SARS-CoV-2 IgG antibodies above 225 AU/mL and an antibody-neutralizing capacity of at least 25% inhibition.

Nucleocapsid (N)-protein antibody detection was used to detect previous SARS-CoV-2 infection. N-protein antibodies were determined by electrochemiluminescence (Anti-SARS-CoV-2 Elecsys^®^ assay, Roche Diagnostics, GmbH, Germany) using a COBAS analyzer system (Roche Diagnostics) according to the manufacturer’s instructions. Patients were considered previously infected with SARS-CoV-2 if N-protein antibodies were detected, as this protein was not included in the formulation of the vaccines administered to the participants.

### 2.3. Statistical Analysis

Patients were matched to controls by age and sex, with a ratio of 1:9, using the cardinality method, with balance tolerability set to 0.1 for both age and sex.

We performed a linear mixed-effects model with repeated measurements to investigate how the change in immune response differed among patients and controls. The primary outcome was IgG antibody levels, which were log-transformed before analyses. Fixed effects were follow-up visits and groups, defined as patient or control. Log-transformed IgG levels were modeled as a function of the visit and the interaction between the group and the visit. We included a random effect for each subject to account for the repeated measurements from the same participant. We assumed an unstructured covariance pattern for the random effects and fitted the model using the Restricted Maximum Likelihood (REML) method. Model assumptions were assessed through residual diagnostics, variance homogeneity checks, and convergence tests, all of which were satisfactory. Singularity checks confirmed an identifiable random-effects structure and a stable model fit.

Continuous data were reported as medians with range and categorical variables as frequencies and percentages. Differences in characteristics were assessed with the Mann–Whitney U test for continuous variables. Fischer’s exact test was used for categorical variables when the counts were of less than five, while the Chi-squared test was applied when the counts exceeded five.

IgG antibody levels and neutralization capacity were reported as the estimated rate ratios from the mixed-effects models. For clarity, we also report the mean difference in percentages derived from each estimated rate ratio. In addition, 95% confidence intervals (CI) and corresponding *p*-values are reported. A *p*-value ≤ 0.05 was considered statistically significant in all analyses. Differences in immune response after COVID-19 vaccination between the patients and controls were tested using Fisher’s exact test.

Sensitivity analyses for age, sex and the number of received vaccinations were conducted by adjusting for each variable separately in the mixed-effects model. Further, we examined the three-way interaction between the number of vaccines and group (patients and controls) and visit.

Statistical analyses were performed in RStudio 4.2.0 using the following packages: “MatchIt”, “dplyr”, “Rglpk”, “LMMstar”, “lme4”, and “lmerTest”. Figures and graphs were created with RStudio 4.2.0 using the packages “ggplot2” and “lattice”.

## 3. Results

### 3.1. Study Population

We included 34 patients, of whom 26 were diagnosed with CIDP and 8 with MMN. A total of 306 age- and sex-matched controls were included. There was a similar distribution of age (*p* = 0.052), body mass index (*p* = 0.170), and sex (*p* = 0.739) in the two groups. Thirty-three patients received immunoglobulin therapy, and one patient received CD20-depleting therapy, Rituximab, during the follow-up period. At the 12-month follow-up, 91% of controls and 62% of patients had received four vaccine doses (*p* < 0.001). Additionally, 3% of controls received a fifth dose after 24 months. In contrast, 59% of patients received a fifth vaccine dose (*p* < 0.001). At baseline, the proportion of participants naturally infected with SARS-CoV-2 was similar, with 7% of controls and 6% of patients being infected. The descriptive characteristics are presented in Table 1.

During follow-up, N-antibody-verified COVID-19 infection was more frequently observed in patients with CIDP and MMN than in controls. SARS-CoV-2 occurred in 47% of controls and 62% of patients; however, the difference was insignificant (*p* = 0.138). For vaccine types across visits, see Appendix A. A detailed per-visit number of N-antibody-confirmed SARS-CoV-2 infections in patients with CIDP and MMN and controls is presented in Appendix A. The proportion of participants with reinfections (defined as ≥2 N-antibody-verified COVID-19 infections) throughout this study is presented in Appendix A.

### 3.2. SARS-CoV-2 Anti-RBD IgG Levels

SARS-CoV-2 anti-RBD IgG levels were compared across visits using a mixed-effects model. Compared to the baseline, we found an increase in IgG levels at all follow-up visits (*p* < 0.001, Figure 1). At baseline, the mean IgG levels were 3 AU/mL in the controls and 5 AU/mL in the patients. After three weeks, the IgG levels increased to 801 AU/mL in controls and 567 AU/mL in patients. By the two-month follow-up, the IgG levels had further risen to 14,884 AU/mL in the controls and 10,730 AU/mL in the patients.

Six months after the first vaccination, the IgG levels declined to 2465 AU/mL in the control group and 1309 AU/mL in the patient group. At the 12-month follow-up, the IgG levels increased to 17,774 AU/mL in controls and 11,896 AU/mL in patients. A decline in the IgG levels was observed at the 18-month follow-up (14,678 AU/mL for controls and 7694 AU/mL for patients), followed by an increase in the IgG levels at the 24-month follow-up (34,647 AU/mL for controls and 13,309 for patients) (Figure 1).

Differences in SARS-CoV-2 anti-RBD IgG levels between controls and patients were examined across visits. No differences between groups were observed at baseline (mean difference 30% [estimated rate ratio 0.70, 95%CI: 0.30–1.64], *p* = 0.41), at the 21-day follow-up (41% [1.41, 95%CI: 0.68–2.93], *p* = 0.35), or at the 3-month follow-up (39% [1.39, 95%CI: 0.88–2.19], *p* = 0.16). At 6 months, follow-up controls had 88% higher IgG levels than patients (88% [1.88, 95%CI: 1.18–3.00], *p* = 0.008). At the 12-month follow-up, no difference was observed (49% [1.49, 95%CI: 0.94–2.37], *p* = 0.089). The control group had higher IgG levels than the patients at the 18-month follow-up (91% [1.91, 95%CI: 1.09–3.33], *p* = 0.023) and at the 24-month follow-up (160% [2.60, 95%CI: 1.73–3.91], *p* < 0.001). Sensitivity analyses for age and sex showed similar results. Adjustment for the number of COVID-19 vaccinations also showed similar results, and no interaction effects were found (all *p* > 0.05).

Sensitivity analyses were performed, excluding the patient who received CD-20-depleting treatment. No difference in SARS-CoV-2 anti-RBD IgG levels was found between patients with MMN and CIDP undergoing immunoglobulin therapy compared to controls, except at the 24-month follow-up, where controls had higher IgG levels (56% [0.44, 95% CI 0.29–0.66], *p* < 0.001).

Adjusting for current COVID-19 infection, defined as the presence of N-antibody-verified COVID-19, increased the mean group difference in IgG levels with significant differences at the 6-, 12-, 18-, and 24-month follow-ups. The infection-adjusted between-group IgG differences are presented in Appendix A.

### 3.3. Neutralization Capacity

The neutralization capacity increased over time compared to the baseline for both groups. At baseline, the neutralization capacity was 8.5% in the controls and 12.7% in the patients. At the 3-week follow-up, the neutralization capacity increased to 38.5% in the control group and 35.0% in the patient group. By the 2-month follow-up, the levels reached 95.2% and 94.0%, respectively. At 6 months, the neutralization capacity declined to 92.0% in the controls and 81.6% in the patients. At 12 months, it increased to 98.0% and 95.8%, followed by 98.8% and 91.5% at 18 months. At the 24-month follow-up, the neutralization capacity was 97.8% in the control group and 89.3% in the patient group (Figure 2). As regards SARS-CoV-2 IgG levels, no differences were found between the two groups at baseline (mean difference −4.2%, 95%CI: −12.12–3.69, *p* = 0.294), after 21 days (3.5%, 95%CI: −9.89–16.97, *p* = 0.604), or at the 3-month follow-up (1.3%, 95%CI −4.72–7.21, *p* = 0.680). The control groups showed a higher neutralization capacity at the 6-month follow-up than patients with CIDP and MMN (10.4%, 95%CI: 3.36–17.40, *p* = 0.004). We observed no difference at the 12-month follow-up (2.2%, 95%CI: −1.86–6.35, *p* = 0.283). The control group had a higher neutralization capacity compared to patients with 18-month follow-up (7.3%, 95%CI: 3.02–11.67, *p* < 0.001) and 24-month follow-up (8.5%, 95%CI: 3.11–13.971, *p* = 0.002) (Figure 2).

Sensitivity analyses were performed, excluding the patient on CD-20-depleting treatment, with similar results. The control group had a higher neutralization capacity at the 6-month follow-up than patients with CIDP and MMN in immunoglobulin therapy (10.87%, 95%CI: 3.72–18.02, *p* = 0.003), at the 18-month follow-up (4.96%, 95%CI: 0.96–8.95, *p* = 0.015), and the 24-month follow-up (5.51%, 95%CI: 0.52–10.49, *p* = 0.031).

Adjusting for current N-antibody-verified COVID-19 infection at sampling time did not affect the differences in the neutralization capacity between the two groups.

### 3.4. Responders to Vaccination

Across all visits conducted during the 24 months following the first vaccine dose, 292 study participants showed a positive immune response, defined as SARS-CoV-2 anti-RBD IgG levels of at least 225 AU/mL and a neutralization capacity of at least 25% at any time point. Participants with previous natural COVID-19 infection were more likely to be vaccine responders (OR 39.4, 95% CI: 10.0–341.8, *p* < 0.001).

One participant had missing values after week 3 and was excluded from further responder analyses. A total of 97.1% of patients with MMN and CIDP (32/33) demonstrated a positive vaccine response within the 24-month follow-up, characterized by SARS-CoV-2 anti-RBD IgG levels of at least 225 AU/mL and a neutralization capacity of at least 25%. The single vaccine non-responder received CD-20-depleting therapy during the follow-up period, whereas all other patients were treated with immunoglobulin. In total, 85% of the controls had had a vaccine response within 24 months (260/306). There was no change in the number of responders between patients and controls after 24 months (95%CI: 0.89–235.25, *p* = 0.064; Figure 3).

However, at the 24-month follow-up visit alone, patients were less likely to be vaccine responders than the controls (*p* = 0.003). No differences in vaccine response were observed between the two groups at any other time point (Table 2).

## 4. Discussion

In this observational cohort study, patients with MMN and CIDP demonstrated short- and long-term immune responses to COVID-19 vaccination comparable to those of controls. All patients receiving immunoglobulin therapy met our definition of vaccine responders. However, our analysis revealed lower IgG antibody levels and lower neutralization capacity at 6, 18, and 24 months after the first vaccination in patients with MMN and CIDP compared to the control group.

During the COVID-19 pandemic, vaccination programs were rapidly implemented before sufficient data were available to determine whether the immune response to the vaccines was adequate across different patient groups. Studies suggest that many patient groups may exhibit a diminished humoral response to vaccination, which could potentially increase their risk of SARS-CoV-2 infection. This includes a study that found an impaired immune response in 362 patients with lymphoid malignancy, measured as low IgG antibody levels and a low neutralization capacity, compared to immunocompetent individuals, both short-term (within one year) and long-term (more than one year) after the first vaccination [16]. Consistent with this, a study examining patients with chronic autoimmune diseases, including multiple sclerosis, inflammatory bowel diseases, and rheumatoid arthritis, who were treated with glucocorticoids or B-cell-depleting medications, demonstrated that these patients had lower SARS-CoV-2 antibody levels two weeks after receiving a COVID-19 vaccination compared to healthy individuals [13]. Another study demonstrated reduced IgG antibody levels in 378 persons with HIV compared to controls at both 3 weeks and 2 months post-vaccination [23]. However, no difference in antibody concentration or cellular response was observed in this cohort 11 months after a third COVID-19 vaccination. A study investigating the humoral immune response after COVID-19 vaccination in patients with myasthenia gravis, an autoimmune disease driven by the production of autoantibodies targeting the postsynaptic acetylcholine receptor in skeletal muscle, found lower SARS-CoV-2 IgG levels at 3 and 12 weeks after vaccination, but 6 months after vaccination, the humoral response was the same in patients and controls [15]. Overall, the findings from these studies indicate that patients undergoing treatment with immunosuppressants exhibit a diminished or delayed immune response compared to immunocompetent individuals. This suggests that these patients may be less protected against SARS-CoV-2 infection due to an impaired response to COVID-19 vaccination.

Only one other study has examined the antibody response following COVID-19 vaccination in patients with immune-mediated polyneuropathies in immunoglobulin treatment. The study reported a reduced short-term antibody response, with lower IgG levels in patients 21 days after the initial vaccination compared to controls [24]. In contrast, we did not observe any difference in the immune response during the initial weeks after vaccination. Notably, the differences in findings may be influenced by the fact that the other study had a very small sample size with only 5 patients at the 21-day follow-up and 4 patients at the 12-week follow-up, and its study conclusion is based on insignificant results.

Our study involves a more extended follow-up period than those of comparable studies investigating the immune response to COVID-19 vaccination in patients treated with immunosuppressive therapy. Therefore, the 24-month follow-up indicating a sufficient long-term immune response to COVID-19 vaccination cannot be directly compared to previous study results. However, the study investigating lymphoid malignancy reported a decrease in immune response 12 months after the first COVID-19 vaccination, whereas the study on patients with myasthenia gravis observed a delayed immune response compared to immunocompetent individuals 6 months post-vaccination. The differences between our study and previous studies do not appear to be related to methodological variations, as the patients originated from the same study cohort and antibody levels were measured using the same technique. Instead, the difference in results between the two studies and ours could be explained by other factors. First, the studies investigate different diseases. Second, the difference in immune responses following COVID-19 vaccination may also relate to the various treatments. Among the 57 patients with myasthenia gravis, most were treated with either azathioprine (51%) or low-dose glucocorticoids (14%) [15]. In contrast, patients with lymphoid malignancies were treated with immunosuppressive therapies such as anti-CD20 therapy or conventional chemotherapy, which are known to be more intense immunosuppressants [16]. Anti-CD20 monoclonal antibodies selectively deplete B cells expressing CD20. The direct actions on the B cells and the indirect effects on other immune pathways make it an efficient immunosuppressant [25]. In immunoglobulin therapy, antibodies are administered either intravenously or subcutaneously [26]. The mechanisms of action of immunoglobulin are not fully understood. Nevertheless, they may both inhibit inflammatory pathways and provide antibodies to fight infections or boost the immune system in immunodeficient individuals [27]. A study examining humoral and T-cell immunity in different fragile groups after COVID-19 vaccination found that treatment type was more capable of predicting the humoral response than disease type [12]. This finding is consistent with our results, as the only vaccine non-responder in our study was undergoing CD-20-depleting therapy during the follow-up period, while all patients receiving immunoglobulin therapy demonstrated a positive immune response to the COVID-19 vaccination. However, as our study only included one patient, we cannot draw clear conclusions regarding patients with CIDP and MMN in CD-20-depleting therapy. It is important to note that the number of vaccines among patients with MMN and CIDP and controls differs. All 21 participants who received a first dose of Astra Zeneca ChAdOx1 were in the control group, with none in the group of patients with MMN and CIDP. Given the small proportion and the combination with mRNA vaccines, we do not expect this to have influenced the results. Our findings indicate that patients with MMN and CIDP have positive short- and long-term immune responses to COVID-19 vaccination. The IgG levels and neutralization capacity were lower compared to those of the controls at 6, 18, and 24 months post-vaccination, which may reflect a more rapid decline in the immune response, indicating a shorter duration of vaccine-induced immunity in patients with MMN and CIDP. However, the levels remained above the threshold for a positive immune response, and the reduced levels may not be clinically relevant. The lower IgG and neutralization capacity levels observed in patients may reflect the contribution of the immunoglobulin therapy as it can affect the measured antibody levels. However, these effects are generally modest and recent studies did not find a reduction in the detection of vaccine-induced antibodies during immunoglobulin treatment if the vaccination is administered at least two weeks before or after [28].

Additionally, we found no difference in the number of responders between patients with CIDP and MMN compared to controls after 24 months. This suggests that while antibody levels and neutralization capacity are reduced in these patients, they still appear to maintain a sufficient immune response, highlighting the potential benefit of COVID-19 vaccination. When looking at the 24-month visit alone, fewer patients had a positive immune response. It is important to notice that natural COVID-19 infection, which was more prevalent in the patient group than the control group at 6-month and 12-month follow-up, can increase IgG and neutralization capacity levels without reflecting vaccine-induced immune response.

While the observed reductions in antibody levels may not translate into clinically meaningful risks, targeted monitoring of patients with additional risk factors, such as those on concurrent CD20-depleting therapy, may be warranted. Furthermore, our results highlight that immunoglobulin therapy does not appear to impair vaccine efficacy, supporting its continued use in these patients during COVID-19 vaccination programs.

Our study is the largest to date investigating the humoral immune response in patients with immune-mediated polyneuropathies undergoing immunoglobulin treatment. A key strength of this study is the long follow-up period of 24 months, making it possible to investigate both short- and long-term humoral immune responses in patients with immune-mediated polyneuropathies in immunoglobulin therapy. Another strength of the study is the data on N-protein antibodies, which makes it possible to investigate previous natural COVID-19 infections.

There are also limitations to this study. A limitation of our study is that antibody levels were measured against the wild-type RBD using a fixed assay cut-off, which may not fully reflect protection against emerging SARS-CoV-2 variants. Although our study is larger than previous studies, the number of patients is still relatively small. Most patients were vaccine responders, and only one patient was a non-responder, which limited our ability to perform sub-analyses of risk factors of vaccine non-response. However, the patient cohort is large enough to detect statistical differences in the long-term humoral immune response, unlike the previous study, which included only four patients. A positive immune response may result from a primary infection rather than vaccination. However, we consider N-antibody-confirmed COVID-19 infection a proxy for natural infection, and no statistical difference was observed between the two groups. Another limitation of our study is the sampling schedule, which may have missed peak immune responses following vaccination, potentially underestimating the maximal antibody levels achieved. Additionally, there are missing data as some patients and controls did not have blood samples taken at all seven planned visits. To account for the missing data, we applied a linear mixed model.

We demonstrate that patients with CIDP and MMN have a positive immune response following COVID-19 vaccination and that all patients treated with immunoglobulin are vaccine responders. While healthcare systems and treatment protocols may vary, the fundamental immunological mechanisms underlying CIDP and MMN are consistent across populations. Therefore, we believe our findings are broadly applicable. More extensive studies are required to identify risk factors associated with non-response to COVID-19 vaccination, including distinguishing the effects of disease pathophysiology and immunosuppressant therapy. This knowledge is important for refining future vaccine strategies. It could help identify vulnerable subgroups among patients with CIDP and MMN who would benefit from booster doses while ensuring that other patients do not receive unnecessary additional vaccinations.

## 5. Conclusions

Patients with CIDP and MMN demonstrated a positive humoral response following COVID-19 vaccination. While the SARS-CoV-2 IgG levels and neutralization capacity were lower in patients compared to those of controls at 6, 18, and 24 months, the proportion of non-responders did not differ between the two groups. All patients receiving immunoglobulin therapy demonstrated a positive vaccine response.

## Figures and Tables

**Figure 1 vaccines-13-00902-f001:**
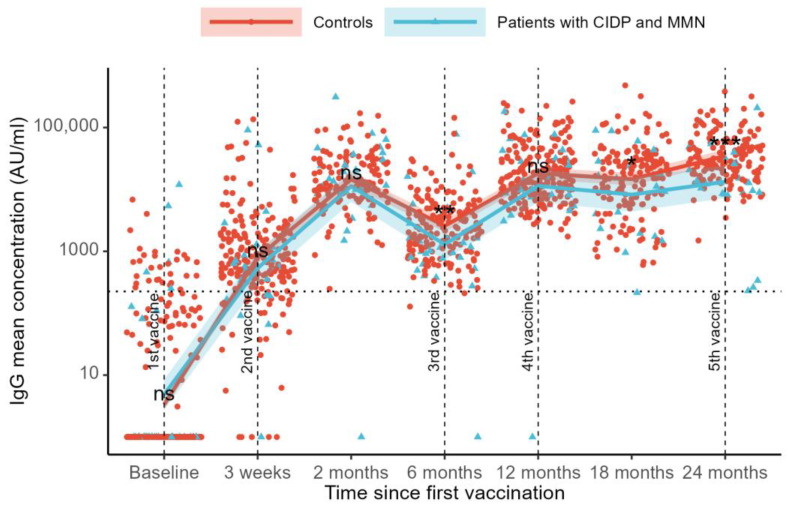
Observed SARS-CoV-2 IgG antibody levels and predicted mean concentration up to 2 years after initial COVID-19 vaccination in patients with CIDP and MMN and controls. The model of the predicted mean, with 25% and 75% confidence intervals, is plotted against the observed individual IgG levels in AU/mL on a log10 scale. Observed data are represented by red circles for controls and blue triangles for patients. Predicted mean concentrations are shown as a red line for controls and a blue line for patients; the lines show back-transformed (exponentiated) model estimates on a log10 *y*-axis. Shaded areas around the lines indicate the confidence intervals for the predictions. The *y*-axis depicts the concentration in AU/mL on a non-logarithmic, back-transformed, scale. The horizontal dotted line represents the lower threshold defined for a positive vaccine response. The vertical dotted lines indicate the times at which vaccinations were administered. Statistical significance of group differences at each visit is represented with stars: * = *p* < 0.05, ** = *p* < 0.01, *** = *p* <0.001, ns = not significant. CIDP: Chronic Inflammatory Demyelinating Polyneuropathy. MMN: Multifocal Motor Neuropathy.

**Figure 2 vaccines-13-00902-f002:**
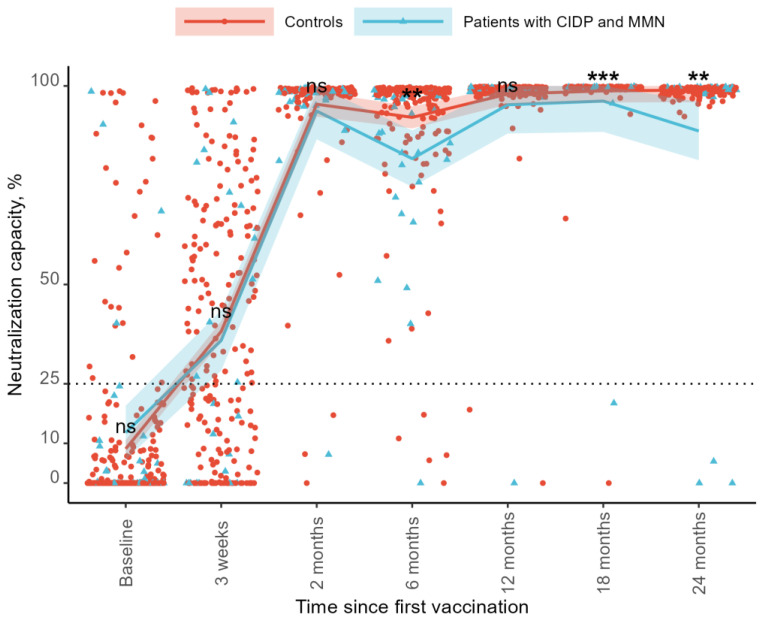
Observed neutralization capacity and predicted neutralization capacity up to 2 years after initial COVID-19 vaccination in patients with CIDP and MMN and controls. The model of the predicted neutralization capacity with 25% and 75% confidence intervals is plotted with the observed individual neutralization capacity in percentages (%). Observed data are represented by red circles for controls and blue triangles for patients. Predicted neutralization capacities are shown as a red line for controls and a blue line for patients. Shaded areas around the lines indicate the confidence intervals for the predictions. The horizontal dotted line represents the lower threshold defined for a vaccine response. Statistical significance of group differences at each visit is represented with stars: ** = *p* < 0.01, *** = *p* <0.001, ns = not significant CIDP: Chronic Inflammatory Demyelinating Polyneuropathy. MMN: Multifocal Motor Neuropathy.

**Figure 3 vaccines-13-00902-f003:**
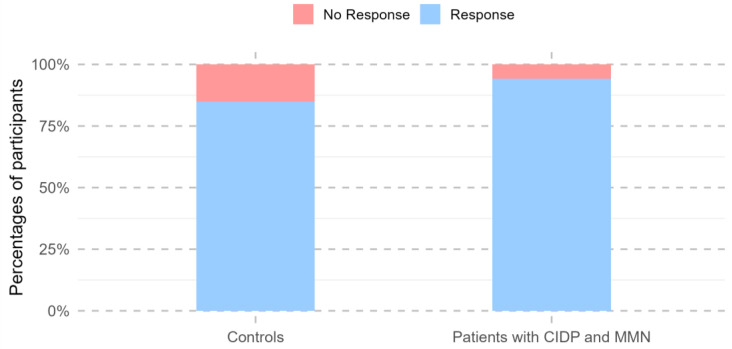
Rates of vaccine responders over the two-year follow-up period after initial COVID-19 vaccination in controls and patients with CIDP and MMN receiving immunoglobulin therapy. The *y*-axis represents the percentage of participants, while the *x*-axis depicts the two groups of participants. Vaccine responders (blue) were defined by SARS-CoV-2 IgG antibodies exceeding 225 AU/mL and an antibody-neutralizing capacity of at least 25% inhibition. Non-responders are shown in red. CIDP: Chronic Inflammatory Demyelinating Polyneuropathy. MMN: Multifocal Motor Neuropathy.

**Table 1 vaccines-13-00902-t001:** Descriptive characteristics of the study population.

		Controls (*n* = 306)	Patients (*n* = 34)	*p*-Value
Age, years median (IQR)		67.0 (63.0–70.0)	69.5 (63.3–74.8)	0.052
Male sex, *n* (%)		184 (60.1%)	22 (64.7%)	0.739
BMI, median (IQR)		25.1 (22.6–27.5)	26.6 (23.2–29.1)	0.170
MMN, *n* (%)			8 (23.5)	
CIDP, *n* (%)			26 (76.5)	
Therapy type, *n* (%)	Immunoglobulin therapy		33 (97.1)	
	CD-20 depleting therapy		1 (2.9)	
No. of vaccines, *n* (%)	1, *n* (%)	306 (100.0)	34 (100.0)	1.000
	2, *n* (%)	303 (99.0)	33 (97.1)	0.345
	3, *n* (%)	282 (92.2)	27 (79.4)	0.033
	4, *n* (%)	279 (91.2)	21 (61.8)	<0.001
	5, *n* (%)	10 (3.3)	20 (58.8)	<0.001
No. of samples available at timepoint, *n* (%)	Baseline	306 (100.0)	34 (100.0)	1.000
	3-week follow-up	221 (72.2)	25 (73.5)	1.000
	2-month follow-up	145 (47.4)	29 (85.3)	<0.001
	6-month follow-up	188 (61.4)	30 (88.2)	0.002
	12-month follow-up	214 (70.0)	28 (82.4)	0.188
	18-month follow-up	167 (54.6)	24 (70.6)	0.042
	24-month follow-up	154 (50.3)	27 (79.4)	0.002
N-antibody-verified COVID-19 infection, *n* (%)	Baseline/before first dose	21 (6.9)	2 (5.9)	1.000
	During Follow-up	143 (46.7)	21 (61.8)	0.138

IQR, interquartile range; *n*, number; BMI, body mass index; MMN, Multifocal Motor Neuropathy; CIDP, Chronic Inflammatory Demyelinating Polyneuropathy.

**Table 2 vaccines-13-00902-t002:** Number of vaccinations administered and vaccine responders across visits for patients with CIDP and MMN and controls.

	Patients with CIDP and MMN			Controls			
Visit	No. of Vaccine Doses, Median (IQR)	Days from Last Vaccine, Median (IQR)	Vaccine Responders, N (%)	No. of Vaccine Doses, Median (IQR)	Days from last Vaccine, Median (IQR)	Vaccine Responders, N (%)	*p*-Value¤
Baseline	0 (0–0)	3.5 (3.0–4.5)	2/32 (6.3)	0 (0–0)	0.0 (0.0–0.0)	18/297 (6.1)	1
21 days	1 (1–1)	20.0 (19.0–22.0)	12/25 (48.0)	1 (1–1)	21.0 (15.0–23.0)	122/221 (55.2)	0.636
3 months	2 (2–2)	33.0 (24.3–40.0)	28/29 (96.6)	2 (2–2)	31.0 (25.0–35.0)	142/145 (97.9)	0.521
6 months	2 (2–2)	141.5 (134.8–156.8)	27/29 (93.1)	2 (2–2)	137.0 (126.8–166.0)	182/188 (96.8)	0.290
12 months	3 (3–3)	111.0 (91.0–129.0)	27/28 (96.4)	3 (3–3)	85.0 (77.0–97.0)	211/213 (99.0)	0.311
18 months	3 (3–3)	315.0 (307.0–331.00)	23/24 (95.8)	3 (3–3)	318.0 (308–325)	166/167 (99.4)	0.236
24 months	4 (4–4)	121.0 (93.3–129.0)	24/27 (88.9)	4 (4–4)	105 (99.0–113.0)	154/154 (100.0)	0.003 *

*p*-value for the difference in vaccine response between the two groups at each visit. * statistically significant. Abbreviations: CIDP; Chronic Inflammatory Demyelinating Polyneuropathy. MMN: Multifocal Motor Neuropathy. IQR: interquartile range.

## Data Availability

The data presented in this study are available in de-identified format on request from the corresponding author due to privacy and ethical reasons.

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
