# Peer review of "Humoral Immune Response Following COVID-19 Vaccination in Multifocal Motor Neuropathy and Chronic Inflammatory Demyelinating Polyneuropathy"

_vaccines, 2025, doi:10.3390/vaccines13090902_

Round 1
Reviewer 1 Report
Comments and Suggestions for Authors
The article is well-written and scientifically sound; however, several important aspects remain unaddressed, and these points should be considered in the revised manuscript.
Lines 51–53: Is the differentiation between MMN and CIDP adequately elucidated, particularly regarding their pathogenesis and clinical presentations? Could additional details be provided regarding the immunological pathways that distinguish MMN from CIDP? If yes, please write a short notes.
Lines 56 to 59: Are there any guidelines cited that designate IVIg as the preferred therapy for MMN and CIDP? Kindly provide justification.
Lines 74 to 76: In what ways do vaccine responses differ among various immunosuppressive medications, such as corticosteroids compared to monoclonal antibodies? Is there any discourse regarding the cellular immune response (T-cell mediated), in addition to humoral immunity, in these patients?
Lines 86–88: What data substantiates the implementation of enhanced immunization procedures in individuals with MMN/CIDP specifically? Are there any safety issues or negative effects observed in this population after many booster vaccinations?
How does the study substantiate the necessity to evaluate vaccine-induced immune responses specifically in MMN and CIDP patients undergoing immunoglobulin therapy, given the lack of prior data? Kindly provide justification.
Were potential confounders such as prior SARS-CoV-2 infection, comorbidities, or other immunosuppressive drugs accounted for in the analysis? The inquiry pertains to the detection of nucleocapsid antibodies and the patient medication history, specifically lines 131–136.
What was the rationale for employing a 1:9 matching ratio, and may this disparity have influenced the statistical power or introduced bias in the comparison of immune responses? As indicated in Lines 138–139
Is the longitudinal sampling schedule (up to 24 months) suitable for capturing both peak and diminishing immune responses, and were there any concerns regarding participant retention or sample completeness as mentioned in Lines 106-108 and 141-146?
Were the ELISA assays for IgG and neutralizing capacity confirmed against international standards, and how was variability in the assays managed over the two-year trial period? Lines 117–126
Was the application of a linear mixed-effects model suitable for assessing repeated assessments of highly varied individual immune responses? Were the model assumptions evaluated and satisfied? Lines 141 to 148
Can the results from this Danish cohort be extrapolated to MMN and CIDP patients in alternative healthcare environments, especially those with varying immunization regimens or IVIg protocols? Lines 98 to 100
Table 1; how do the disparities in vaccine doses administered to patients versus controls influence the analysis of immune response data, especially given that a markedly greater percentage of patients received a fifth dose (59% vs 3%, p < 0.001), while a smaller proportion received only 3 or 4 doses compared to controls?
Figure 1: Considering the persistently diminished IgG levels in patients with CIDP and MMN relative to healthy controls at all intervals (notwithstanding analogous vaccination regimens), how could immunoglobulin therapy or immune dysfunction in these patients have impacted their capacity to elicit and sustain vaccine-induced humoral immunity, especially regarding the waning and boosting phenomena observed over 24 months? Please justify !!
What accounts for the statistical significance of variations in anti-RBD IgG levels between controls and patients emerging only after the 6-month follow-up, despite comparable initial responses and vaccination regimens?
To what degree could prolonged immunoglobulin therapy in CIDP/MMN patients disrupt the natural synthesis or longevity of SARS-CoV-2-specific antibodies after several vaccinations?
Despite the sustained high neutralization capacity in both cohorts, what are the possible therapeutic ramifications of the minor yet statistically significant variations in neutralization capacity noted at 6, 18, and 24 months?
Considering that the elimination of the sole CD-20-depleting treatment patient did not affect the primary results, how resilient are the identified group differences to other possible outliers or the impact of a limited sample size, particularly at extended follow-up intervals?
Section 3.4: What immunological or clinical characteristics may elucidate the reason a greater percentage of CIDP/MMN patients (97.1%) responded to vaccination in contrast to healthy controls (85%), despite the patients receiving immunoglobulin therapy?
Despite the p-value at 24 months (p = 0.064) lacking statistical significance, does the extensive confidence interval (95% CI 0.89–235.25) imply that the sample size was inadequate to identify a substantial difference in long-term responder status?
Table 2: What accounts for the increased median number of vaccine doses received by patients with CIDP/MMN at 24 months compared to controls, despite a decrease in the proportion of vaccination responders at the final assessment? Could this indicate a qualitative distinction in immunological memory or the necessity for various dosage regimens in these patients?
Discussion:
Despite the reduced IgG levels and neutralization capacity observed in CIDP and MMN patients at 6, 18, and 24 months, these levels remained above the threshold for response. What is the therapeutic significance of these reductions for true protection against SARS-CoV-2 infection and severe disease?
Given the identification of a single vaccine non-responder (undergoing CD20-depleting medication), how could these data guide tailored vaccine booster regimens for patients with CIDP and MMN, particularly those receiving more potent immunosuppressants?
The study cites variations in immune responses among disorders such as myasthenia gravis and lymphoid malignancies. How can future research determine whether reduced responses are primarily influenced by disease pathophysiology or the nature/intensity of immunosuppressive therapy?
Author Response
Author responses to reviewer 1 are seen below. For all point-by-point responses to the reviewer's comments, please see the attachment.
Reviewer 1, comment 1:
The article is well-written and scientifically sound; however, several important aspects remain unaddressed, and these points should be considered in the revised manuscript.
Response 1: Thank you for your constructive feedback. We appreciate your recognition of the scientific quality of our work. Below, we address each of the reviewer’s comments and concerns point by point.
Reviewer 1, comment 2:
Lines 51–53: Is the differentiation between MMN and CIDP adequately elucidated, particularly regarding their pathogenesis and clinical presentations? Could additional details be provided regarding the immunological pathways that distinguish MMN from CIDP? If yes, please write a short notes.
Response 2: CIDP and MMN are both immune-mediated neuropathies. Their exact pathogenesis is not fully understood. CIDP involves different immunopathological mechanisms, including cellular, humoral, and complement-mediated pathways. MMN also involves different immunological mechanisms, with 30–50% of patients having IgM anti-GM1 autoantibodies. The two disorders differ in their clinical presentation. CIDP affects both motor and sensory nerves, while MMN is limited to motor involvement. This has been clarified in the introduction, lines 52-56, with references to relevant literature.
Reviewer 1, comment 3:
Lines 56 to 59: Are there any guidelines cited that designate IVIg as the preferred therapy for MMN and CIDP? Kindly provide justification.
Response 3: Thank you for your comment. Both immunoglobulins and corticosteroids are recommended treatments for CIDP. The choice between intravenous (IVIg) and subcutaneous (SCIg) immunoglobulin does not affect treatment efficacy and is typically guided by national preferences based on cost and administration benefits. Corticosteroids are generally cheaper but associated with more long-term side effects Treatment recommendations can vary by country. In Denmark, immunoglobulin is preferred. Immunoglobulin is the only preferred therapy for MMN. This has now been clarified in the introduction, lines 60-70, with references to international and national guidelines.
Reviewer 1, comment 4:
Lines 74 to 76: In what ways do vaccine responses differ among various immunosuppressive medications, such as corticosteroids compared to monoclonal antibodies? Is there any discourse regarding the cellular immune response (T-cell mediated), in addition to humoral immunity, in these patients?
Response 4: Thank you for this relevant comment. While our study focused on humoral immune responses, we acknowledge that different types of immunosuppressive therapies may impact immune responses to vaccination in different ways. For example, corticosteroids suppress both humoral and cellular immunity while B-cell-depleting monoclonal antibodies primarily impair the humoral immune response. We have added a sentence in the introduction clarifying the different types of immunosuppressants and their mechanisms and a sentence in the discussion highlighting the importance of further studies evaluating T-cell–mediated immunity in this patient population.
Reviewer 1, comment 5:
Lines 86–88: What data substantiates the implementation of enhanced immunization procedures in individuals with MMN/CIDP specifically? Are there any safety issues or negative effects observed in this population after many booster vaccinations?
Response 5: Thank you for this comment. As stated in the introduction, the implementation of enhanced vaccine frequency in patients with CIDP and MMN was based on a theoretical rationale due to their use of immunomodulatory or immunosuppressive therapies. At the time, no disease-specific evidence supported this approach. However, some patients with CIDP/MMN were observed to develop severe COVID-19 during the early phases of the pandemic which supported a defensive vaccination strategy. Additionally, although some patients reported transient worsening of CIDP or MMN symptoms following vaccination. We have clarified these points in the revised manuscript, lines 94-107.
Reviewer 1, comment 6:
How does the study substantiate the necessity to evaluate vaccine-induced immune responses specifically in MMN and CIDP patients undergoing immunoglobulin therapy, given the lack of prior data? Kindly provide justification.
Response 6: Thank you for this important question. At the time of study initiation, no data were available on vaccine-induced immune responses in patients with MMN or CIDP. These patients were considered potentially at risk of reduced vaccine efficacy due the immunosuppressant treatment. Therefore, we found it important to investigate whether this population demonstrated a sufficient humoral immune response. We have clarified this in the revised introduction, lines 103-107: “These patients are considered at risk of reduced vaccine efficacy and are offered booster vaccines. On the other hand, vaccination can in rare cases induce worsening of CIDP and MMN symptoms. Therefore, it is important to investigate whether this population demonstrates a sufficient humoral immune response.”
Reviewer 1, comment 7:
Were potential confounders such as prior SARS-CoV-2 infection, comorbidities, or other immunosuppressive drugs accounted for in the analysis? The inquiry pertains to the detection of nucleocapsid antibodies and the patient medication history, specifically lines 131–136.
Response 7: Thank you for this comment. We have addressed prior SARS-CoV-2 infection by adding a Supplementary Table 2, which presents nucleocapsid antibody results (indicative of prior infection) for each visit and group. In addition, we performed adjusted mixed-model analyses controlling for infection status at the time of sampling, and these results were consistent with the main analyses. None of the patients were receiving other immunosuppressive drugs during the study period.
Reviewer 1, comment 8:
What was the rationale for employing a 1:9 matching ratio, and may this disparity have influenced the statistical power or introduced bias in the comparison of immune responses? As indicated in Lines 138–139
Response 8: Thank you for this relevant comment. The 1:9 matching ratio was chosen to take full advantage of the available control data and to maximize statistical power. While different matching ratios can introduce bias, increasing the number of controls per case generally does not compromise validity if matching is appropriately performed. To prevent bias, we have checked the model assumptions by inspecting residual plots, evaluating variance homogeneity, and testing model convergence, and all assumptions were met. Variance structures were examined using plots of residuals and fitted values. Both residual and random-effects diagnostics were acceptable. Further, singularity checks were negative, indicating that the random-effect structure is identifiable, and the fit is stable. This has been clarified in the Methods, lines 170-172.
Reviewer 1, comment 9:
Is the longitudinal sampling schedule (up to 24 months) suitable for capturing both peak and diminishing immune responses, and were there any concerns regarding participant retention or sample completeness as mentioned in Lines 106-108 and 141-146?
Response 9: Thank you for this thoughtful comment. The longitudinal sampling schedule was designed to capture both peak antibody responses shortly after vaccination (after 21 days and 3 months) and longer-term waning over a 24-month period. We do not capture the peak after the third dose as it was given between the six-month and 12-month visit. This would be very interesting; however, our study is an observational study with no impact on when participants receive vaccination.
There was missing data for some time points, but most patients contributed at all time points. Based on Table 1, more than 70% of the patients participated at any of the seven timepoints, and approximately 80% at 24-month follow-up. To account for the missing data, we applied a linear mixed model. We have clarified this in the revised manuscript and rephrased the discussion to highlight the limitation of not capturing all post-vaccination peaks, lines 449-452: “Another limitation of our study is the sampling schedule, which may have missed peak immune responses following vaccination, potentially underestimating the maximal antibody levels achieved.”
Reviewer 1, comment 10:
Were the ELISA assays for IgG and neutralizing capacity confirmed against international standards, and how was variability in the assays managed over the two-year trial period? Lines 117–126
Response 10: We acknowledge the reviewer for highlighting this concern. Several studies have used the international standards to report the antibody levels. However, our study started before the international standard was available. Therefore, we prioritized the consistency within our study by using the same standards over the two-year period as we do not directly compare antibody levels with other reports, only within our, study using the same assay. Moreover, other studies have reported that, despite the lack of adherence to international standards, the differences between the methods used across studies persisted.
(DOI: 10.3389/fmicb.2022.893801 and DOI: 10.1016/j.intimp.2021.108095). Regarding the variability, we have used the same recombinant antibodies as calibrators over the study, validating the different batches that were employed. Moreover, a positive and negative control were included in each ELISA plate, providing robustness to the assay. We have added a short paragraph on the consistency and variability of the ELISA assay in the methods section, lines 143-148.
Reviewer 1, comment 11:
Was the application of a linear mixed-effects model suitable for assessing repeated assessments of highly varied individual immune responses? Were the model assumptions evaluated and satisfied? Lines 141 to 148
Response 11: Thank you for this relevant question. The application of a linear mixed-effects model was appropriate, as it accounts for both fixed effects (e.g., group, time) and random effects (e.g., individual variability), allowing for the analysis of repeated measurements while accommodating the heterogeneity of immune responses across individuals. We evaluated the model assumptions by inspecting residual plots, evaluating variance homogeneity, and testing model convergence, and all assumptions were met (for further details see Reviewer 2, comment 1). We have clarified this in the Methods section.
Reviewer 1, comment 12:
Can the results from this Danish cohort be extrapolated to MMN and CIDP patients in alternative healthcare environments, especially those with varying immunization regimens or IVIg protocols? Lines 98 to 100
Response 12: Thank you for the question. While healthcare systems and treatment protocols may vary, the fundamental immunological mechanisms underlying CIDP and MMN are consistent across populations. Therefore, we believe our findings are broadly applicable, especially regarding the long-term humoral response to COVID-19 vaccination. This has been clarified in the discussion.
Reviewer 1, comment 13:
Table 1; how do the disparities in vaccine doses administered to patients versus controls influence the analysis of immune response data, especially given that a markedly greater percentage of patients received a fifth dose (59% vs 3%, p < 0.001), while a smaller proportion received only 3 or 4 doses compared to controls?
Response 13: We thank the reviewer for this very relevant question and for noting the imbalance in the number of vaccine doses administered between patients and controls. To evaluate whether this influenced our results, we repeated the mixed-model analyses and adjusted for the number of COVID-19 vaccine doses (as a continuous variable). Further, we examined its three-way interactions with group and visit. Adjusting for the number of doses did not change the estimated group–visit differences in anti-RBD IgG levels or neutralizing capacity, and no interaction effects of the number of vaccine doses reached statistical significance (all p > 0.05). These findings indicate that the observed between-group differences do not depend on the number of vaccinations received, and our results cannot be explained by disparities in the number of vaccine doses. We have added this to the Methods and Results section.
Reviewer 1, comment 14:
Figure 1: Considering the persistently diminished IgG levels in patients with CIDP and MMN relative to healthy controls at all intervals (notwithstanding analogous vaccination regimens), how could immunoglobulin therapy or immune dysfunction in these patients have impacted their capacity to elicit and sustain vaccine-induced humoral immunity, especially regarding the waning and boosting phenomena observed over 24 months? Please justify!!
Response 14: We appreciate the reviewer’s question. The persistently lower IgG levels observed in patients may reflect contributions from both IVIG treatment and underlying immune dysfunction. IVIG provides passive antibodies that could transiently affect measured antibody levels and, through its known immunomodulatory effects—such as modulation of B- and T-cell function and induction of regulatory pathways—could theoretically dampen vaccine-induced immune responses. However, these effects are generally modest (see response to comment 16 for further details) and their clinical impact on long-term vaccine immunity remains uncertain. Conversely, immune dysregulation inherent to CIDP and MMN may impair the humoral responses and explain a part of the observed quantitative differences between patients and controls. Importantly, IgG levels remained above detection thresholds and all patients on IVIG mounted a vaccine response, underscoring that both factors may play a role without clear evidence favoring one mechanism over the other. Further studies are needed to understand their relative impact and clinical implications. These important aspects have been addressed in the Discussion, lines 412 - 417 and 461 – 462.
Reviewer 1, comment 15:
What accounts for the statistical significance of variations in anti-RBD IgG levels between controls and patients emerging only after the 6-month follow-up, despite comparable initial responses and vaccination regimens?
Response 15: Although initial antibody responses were comparable between patients and controls, the statistically significant differences at the six-month follow-up likely reflect a more rapid decline in anti-RBD IgG levels among the patients, indicating a shorter duration of vaccine-induced immunity in this population. This is consistent with findings in other immune-mediated conditions, such as myasthenia gravis. We have clarified this in the discussion lines 407-410: “IgG levels and neutralization capacity were lower compared to the controls at 6-, 18-, and 24 months post-vaccination, which may reflect a more rapid decline in the immune response, indicating a shorter duration of vaccine-induced immunity in patients with MMN and CIDP”.
Reviewer 1, comment 16:
To what degree could prolonged immunoglobulin therapy in CIDP/MMN patients disrupt the natural synthesis or longevity of SARS-CoV-2-specific antibodies after several vaccinations?
Response 16: Thank you for this relevant question. Current evidence indicates that prolonged immunoglobulin therapy does not necessarily impair the SARS-CoV-2 IgG antibody response in patients with CIDP or MMN, if the vaccination is administered at least two weeks before or after an IVIg infusion, as recommended. A prospective study in patients receiving IVIg or SCIg found no reduction in vaccine-induced antibody responses under these conditions (DOI: 10.1111/ene.15508; DOI: 10.1111/ene.16079). We have added a brief paragraph in the discussion, lines 412 – 417, to address this point and cited the relevant literature. The study patients have all received vaccinations according to guidelines.
Reviewer 1, comment 17:
Despite the sustained high neutralization capacity in both cohorts, what are the possible therapeutic ramifications of the minor yet statistically significant variations in neutralization capacity noted at 6, 18, and 24 months?
Response 17: We appreciate this insightful comment. The neutralization capacity in our study was expressed as a percentage, which leads to a saturation effect at higher values and may limit the ability to detect subtle differences between groups at specific time points. Consequently, while the statistical analysis indicated minor but significant variations at 6, 18, and 24 months, the clinical interpretation of these findings should be made with caution. In line with your comment and Reviewer 2’s comment 6, we have revised the conclusion to avoid overstating our findings and interpretations. For example, we have rephrased the conclusion from
“Patients with CIDP and MMN had a sufficient humoral response…” to “Patients with CIDP and MMN demonstrated a positive humoral response…” and removed the sentence “This suggests that despite an initial diminished immune response, the booster vaccination provided a robust and sufficient long-term immune response”.
Reviewer 1, comment 18:
Considering that the elimination of the sole CD-20-depleting treatment patient did not affect the primary results, how resilient are the identified group differences to other possible outliers or the impact of a limited sample size, particularly at extended follow-up intervals?
Response 18: Thank you for raising robustness and small-sample concerns. Beyond the analysis excluding the CD20-depleted patient, we repeated the primary analysis after removing other identified outliers. The between-group differences were unchanged. We acknowledge that the smaller sample size at extended follow-up reduces precision, which is reflected in the wider confidence intervals, and have noted this limitation in the Discussion.
Reviewer 1, comment 19:
Section 3.4: What immunological or clinical characteristics may elucidate the reason a greater percentage of CIDP/MMN patients (97.1%) responded to vaccination in contrast to healthy controls (85%), despite the patients receiving immunoglobulin therapy?
Response 19: While a greater proportion of CIDP/MMN patients (97.1%) responded to vaccination compared to healthy controls (85%), this difference did not reach statistical significance and should therefore be interpreted with caution. Possible explanations include random variation or differences in prior exposure to SARS-CoV-2, as a higher proportion of patients had evidence of previous infection (Table 1, N-antibody–verified COVID-19).
Reviewer 1, comment 20:
Despite the p-value at 24 months (p = 0.064) lacking statistical significance, does the extensive confidence interval (95% CI 0.89–235.25) imply that the sample size was inadequate to identify a substantial difference in long-term responder status?
Response 20: Thank you for this relevant comment. The wide confidence interval at 24 months reflects the reduced sample size and greater variability at this time point, leading to lower precision of the estimate. It cannot be excluded that the study was underpowered to detect a true difference in long-term responder status. We have clarified this limitation in the Discussion, noting that although the point estimate suggests a possible group difference, the imprecision means the result should be interpreted cautiously.
Reviewer 1, comment 21:
Table 2: What accounts for the increased median number of vaccine doses received by patients with CIDP/MMN at 24 months compared to controls, despite a decrease in the proportion of vaccination responders at the final assessment? Could this indicate a qualitative distinction in immunological memory or the necessity for various dosage regimens in these patients?
Response 21: The increased median number of vaccine doses in patients with CIDP/MMN probably reflects adherence to Denmark’s national guidelines, which recommended additional vaccinations for immunosuppressed individuals but not for the general population. This policy difference, rather than an underlying immunological distinction, is the most plausible explanation for the observed disparity in dosing. As there is currently no universally established threshold defining “true protection”, our results cannot answer this question.
Discussion:
Reviewer 1, comment 22:
Despite the reduced IgG levels and neutralization capacity observed in CIDP and MMN patients at 6, 18, and 24 months, these levels remained above the threshold for response. What is the therapeutic significance of these reductions for true protection against SARS-CoV-2 infection and severe disease?
Response 22: We appreciate this important question. As there is currently no universally accepted threshold that defines “true protection” against SARS-CoV-2 infection or severe disease, our results cannot determine the clinical significance of the observed reductions in IgG levels and neutralization capacity. In our study, the term “adequate response” referred to the assay’s positivity threshold for IgG against RBD, which does not necessarily equate to clinical protection. To avoid potential misinterpretation, we have revised the terminology to “positive response” throughout the manuscript.
Reviewer 1, comment 23:
Given the identification of a single vaccine non-responder (undergoing CD20-depleting medication), how could these data guide tailored vaccine booster regimens for patients with CIDP and MMN, particularly those receiving more potent immunosuppressants?
Response 23: As only one non-responder undergoing CD20-depleting therapy was identified, the finding does not allow for definitive conclusions. However, it aligns with existing evidence from other studies indicating impaired vaccine responses in other patient groups receiving CD20-targeted immunosuppression. This observation highlights the need for further investigation and may support the inclusion of tailored booster strategies for such patients. We have clarified this point in the discussion, lines 402-404.
Reviewer 1, comment 24:
The study cites variations in immune responses among disorders such as myasthenia gravis and lymphoid malignancies. How can future research determine whether reduced responses are primarily influenced by disease pathophysiology or the nature/intensity of immunosuppressive therapy?
Response 24: We appreciate this thoughtful question. Distinguishing the effects of disease pathophysiology from those of immunosuppressive therapy will require dedicated studies designed to control for treatment exposure. Such studies could include treatment-naïve patient cohorts, longitudinal monitoring before and after initiation of immunosuppressive therapy, and cross-disease comparisons stratified by treatment type and intensity. While our current dataset does not allow us to make this distinction, we have elaborated on this point in the discussion as an area for future research, lines 459-462.
Reviewer 2 Report
Comments and Suggestions for Authors
Overall
This manuscript addresses an important clinical question, whether patients with CIDP and MMN who receive long-term Ig therapy mount adequate short- and long-term humoral responses to COVID19 vaccination. The prospective 24-month follow-up, age/sex-matched control cohort, and repeated measurement of both anti-RBD IGG and neutralization represent clear strengths that will interest readers. However, the current version requires several revisions to improve the manuscript.
Major Comments
- The use of cardinality matching with a 9:1 control-to-patient ratio is reasonable for maximizing covariate balance. However, the manuscript does not discuss whether this substantial group size gap was addressed was addressed in the mixed-effects modeling assumptions, such as variance homogeneity, model stability, or weighting. Please clarify or additionally discuss whether the modeling approach sufficiently accounts for this imbalance or consider additional diagnostics or sensitivity analysis to support the robustness of the results.
- The manuscript addressed several group-by-timepoint comparison, such as IgG levels between patients and control in figure 1 and vaccine responder rates over times in figure 2 using individual p-values. However, it appears that no correction for multiple testing was applied. Given the number of comparisons across timepoints and groups, there is a risk of inflated type I error. To improve the statistical rigor, please apply appropriate multiple comparison adjustment methods, such as FDR correction or Tukey’s test. Based on that, please indicate again whether the key findings remain significant after adjustment. If such correction was performed already but just not addressed in the manuscript, please clarify this in the Method section.
- In figure 1, the legend states that values are displayed on a non-logarithmic scale, while the main text indicates log-transformed data were used in modeling. To avoid confusion, please clarify the axis label or consider annotating whether values shown are back-transformed estimates. That would help reader interpretation.
Minor Comments
- The definition of an “adequate response” as anti-RBD IgG ≥ 225 AU/ml appears to be based on early pandemic data (ref [17]). Since the study period spans 2021 to 2023, covering major variant shifts, it may be helpful to briefly clarify the rationale for applying this fixed cut-off throughout, particularly given evolving viral antigenicity. If available, referencing an BAU/ml conversion or referring additional references to explain why the authors set the cut-off value as 225 AU/ml would strengthen interpretability, since this cut-off value is the key metric for conclusion from figure 2.
- Please review references in the overall manuscript. For example, references 1 and 4 are the same reference.
- If possible, please rephrase the discussion and conclusion. Some parts seem like overstated.
Author Response
Author responses to reviewer 2 are seen below. For all author responses to the reviewers, please see the attachment.
Reviewer 2 comment: This manuscript addresses an important clinical question, whether patients with CIDP and MMN who receive long-term Ig therapy mount adequate short- and long-term humoral responses to COVID19 vaccination. The prospective 24-month follow-up, age/sex-matched control cohort, and repeated measurement of both anti-RBD IGG and neutralization represent clear strengths that will interest readers. However, the current version requires several revisions to improve the manuscript.
Response: Thank you for your constructive feedback. We appreciate your recognition of the research question and the strengths of our study design. Below, we address each of the reviewer’s comments and concerns point by point. Below, we have addressed all comments to further strengthen the manuscript.
Major Comments
Reviewer 2, comment 1:
The use of cardinality matching with a 9:1 control-to-patient ratio is reasonable for maximizing covariate balance. However, the manuscript does not discuss whether this substantial group size gap was addressed in the mixed-effects modeling assumptions, such as variance homogeneity, model stability, or weighting. Please clarify or additionally discuss whether the modeling approach sufficiently accounts for this imbalance or consider additional diagnostics or sensitivity analysis to support the robustness of the results.
Response 1:
Thank you for this relevant comment. We agree that it is important to address the difference in group size and to assess the model assumptions. We have checked the model assumptions by inspecting residual plots, evaluating variance homogeneity, and testing model convergence, and all assumptions were met. Variance structures were examined using plots of residuals and fitted values. Both residual and random-effects diagnostics were acceptable. Further, singularity checks were negative, indicating that the random-effect structure is identifiable, and the fit is stable. This has been clarified in the Methods, lines 170-172.
Reviewer 2, comment 2:
The manuscript addressed several group-by-timepoint comparison, such as IgG levels between patients and control in figure 1 and vaccine responder rates over times in figure 2 using individual p-values. However, it appears that no correction for multiple testing was applied. Given the number of comparisons across timepoints and groups, there is a risk of inflated type I error. To improve the statistical rigor, please apply appropriate multiple comparison adjustment methods, such as FDR correction or Tukey’s test. Based on that, please indicate again whether the key findings remain significant after adjustment. If such correction was performed already but just not addressed in the manuscript, please clarify this in the Method section.
Response 2: The primary analyses were conducted using linear mixed-effects models for repeated measures. This analysis accounts for intra-personal observations over time at specific time points and provides overall estimates of the groups, time points, and interaction effects. It is not necessary to correct for multiple testing as the model with repeated measurements inherently accounts for some degree of multiple comparison through the structure and assumptions. We agree that it can be necessary to adjust for multiple testing if multiple analyses are performed. However, that is not the case in our analyses as Figure 1 and Figure 2 each depicts a single linear mixed-effects model.
Reviewer 2, comment 3:
In figure 1, the legend states that values are displayed on a non-logarithmic scale, while the main text indicates log-transformed data were used in modeling. To avoid confusion, please clarify the axis label or consider annotating whether values shown are back-transformed estimates. That would help reader interpretation.
Response 3: Thank you for pointing this out. Our mixed-effects model was fitted on log-transformed outcomes. For visualization, Figure 1 displays the corresponding back-transformed (exponentiated) model estimates in AU/mL. We have improved Figure 1 and clarified this in the figure legend to avoid confusion.
Minor Comments
Reviewer 2, comment 4:
The definition of an “adequate response” as anti-RBD IgG ≥ 225 AU/ml appears to be based on early pandemic data (ref [17]). Since the study period spans 2021 to 2023, covering major variant shifts, it may be helpful to briefly clarify the rationale for applying this fixed cut-off throughout, particularly given evolving viral antigenicity. If available, referencing an BAU/ml conversion or referring additional references to explain why the authors set the cut-off value as 225 AU/ml would strengthen interpretability, since this cut-off value is the key metric for conclusion from figure 2.
Response 4: We understand the reviewer’s concern. We acknowledge the definition of “adequate response” might introduce confusion to the reader as this value is the positive threshold of the assay for IgG against RBD. Therefore, we have modified the term “adequate response” for “positive response” to avoid misinterpretation of our results. As we have used the same SARS-COV-2 antigen all over the study (wildtype RBD), the cut-off remains unchanged as the same assay has been used during this period. As the reviewer mentioned, the virus has shift to diverse variants. Nevertheless, in this study we report the antibody levels against the RBD wildtype; therefore, we have included this as a limitation of the study, lines 408-410. As we explain for Reviewer 1 (see comment 10 from Reviewer 1) the conversion to BAU/ml does not always allow direct comparison between studies. Since we compare longitudinally using the same assay, we considered not necessary the use of BAU/ml conversion in this study. The selected cut-off provides with a sensitivity of 94.3% and a specific of 99.5% (and thus kept unchanged) the presence of IgG against RBD after vaccination and/or infection.
Reviewer 2, comment 5:
Please review references in the overall manuscript. For example, references 1 and 4 are the same reference.
Response 5: Thank you for pointing this out. We have thoroughly reviewed all references in the manuscript and removed any duplicates, including the overlap between references 1 and 4.
Reviewer 2, comment 6:
If possible, please rephrase the discussion and conclusion. Some parts seem like overstated.
Response 6: Thank you for this comment. We have revised the discussion and conclusion to ensure the wording is balanced and that we do not overstate the findings. We have, among other things, rephrased the conclusion from “Patients with CIDP and MMN had a sufficient humoral response…” to “Patients with CIDP and MMN demonstrated a positive humoral response…”, removed the sentence “This suggests that despite an initial diminished immune response, the booster vaccination provided a robust and sufficient long-term immune response”, and refrased the sentence “All patients with MMN and CIDP undergoing immunoglobulin therapy were vaccine responders…” to “All patients receiving immunoglobulin therapy met our definition of vaccine responders.”
Reviewer 3 Report
Comments and Suggestions for Authors
The authors investigated the immune response to COVID-19 vaccination in patients with Multifocal Motor Neuropathy (MMN) and Chronic Inflammatory Demyelinating Polyneuropathy (CIDP), most of all treated with immunoglobulin therapy. They found that while patients with MMN and CIDP did show an adequate humoral response to the vaccine, their SARS-CoV-2 IgG antibody levels and neutralization capacity were generally lower than those in a control group. However, all patients receiving immunoglobulin therapy demonstrated an effective vaccine response. I think the paper should add some information in the field of vaccine response of patients undergoing immune-regulating therapies, however it could be improved with further information mainly regarding the effectiveness in terms of protection from infections in those patients. Moreover some other aspect should be clarified to this reviewer.
I have several concerns that I outline below.
- The authors state, "During follow-up, N-antibody-verified COVID-19 infection was more frequently observed in patients with CIDP and MMN than in controls." This confirms that natural infections occurred during the study, which could influence IgG levels and consequently the "responder" rate. The authors should acknowledge and discuss this potential bias, perhaps by revising the cohorts and considering that this might correlate with lower neutralizing antibody levels in patients compared to controls. Furthermore, the number of infections that occurred during the study should be included in the patient descriptions, as it is known that immunosuppressed patients can experience multiple reinfections. This information is also crucial for evaluating vaccination effectiveness and is directly related to the presence of neutralizing antibodies.
- What about the severity of infections observed among the groups? The authors should include this consideration in the text.
- The inclusion of one patient treated with anti-CD20 in the cohort is not informative within the context of this study. I suggest excluding this case from the study population and perhaps mentioning it in the discussion if deemed relevant.
- The criteria used to define an "adequate response" are only referenced from previous studies. I believe this is an important point that requires a more detailed explanation within the text.
- We can observe in the supplementary table that only a small number of participants were vaccinated with AstraZeneca ChAdOx1. How are these cases distributed among the three groups (MMN, CIDP, and controls)?
- Figures 1 and 2 should include indicators of statistical significance, and their legends should be modified accordingly.
Author Response
Author responses to Reviewer 3 are seen below. For all author responses to the reviewer's comments, please see the attachment.
Reviewer 3
The authors investigated the immune response to COVID-19 vaccination in patients with Multifocal Motor Neuropathy (MMN) and Chronic Inflammatory Demyelinating Polyneuropathy (CIDP), most of all treated with immunoglobulin therapy. They found that while patients with MMN and CIDP did show an adequate humoral response to the vaccine, their SARS-CoV-2 IgG antibody levels and neutralization capacity were generally lower than those in a control group. However, all patients receiving immunoglobulin therapy demonstrated an effective vaccine response. I think the paper should add some information in the field of vaccine response of patients undergoing immune-regulating therapies, however it could be improved with further information mainly regarding the effectiveness in terms of protection from infections in those patients. Moreover some other aspect should be clarified to this reviewer.
I have several concerns that I outline below.
Response: Thank you for recognizing the relevance of our findings and for your suggestions to strengthen the manuscript. We have addressed all your comments point-by-point below.
Reviewer 3, comment 1:
The authors state, "During follow-up, N-antibody-verified COVID-19 infection was more frequently observed in patients with CIDP and MMN than in controls." This confirms that natural infections occurred during the study, which could influence IgG levels and consequently the "responder" rate. The authors should acknowledge and discuss this potential bias, perhaps by revising the cohorts and considering that this might correlate with lower neutralizing antibody levels in patients compared to controls. Furthermore, the number of infections that occurred during the study should be included in the patient descriptions, as it is known that immunosuppressed patients can experience multiple reinfections. This information is also crucial for evaluating vaccination effectiveness and is directly related to the presence of neutralizing antibodies.
Response 1: We agree with the reviewer that SARS-CoV-2 infections during follow-up could influence anti-RBD IgG levels and, consequently, the proportion of vaccine “responders”. Natural infection can increase both IgG and neutralization capacity levels without reflecting vaccine-induced immune responses. To address this, we have included detailed per-visit numbers of N-antibody–confirmed SARS-CoV-2 infections in patients with CIDP and MMN and in controls (Supplementary Table S2A), as well as the proportion of participants with reinfections (≥2 infections) throughout the study (Supplementary Table S2C).
Further, we have adjusted the mixed-model analyses for current N-antibody-verified infection at time of sampling. Adjusting for infection at sampling time increased the mean difference in IgG levels between the two groups, with significant differences at not only 6-, 18-, and 24-month follow-up, but also for 12 month-follow-up. The infection-adjusted between-group IgG differences are presented in Supplementary Table S2B. Adjusting for infection at sampling time did not affect the significant differences in neutralization capacity. We have revised Results (lines 258-261 and 284-285 + Supplementary table S2B) and Discussion to acknowledge that natural infection during follow-up may have increased IgG and neutralization capacity and could affect responder rates (lines 425-428).
Reviewer 3, comment 2:
What about the severity of infections observed among the groups? The authors should include this consideration in the text.
Response 2: We agree that the severity of SARS-CoV-2 infections would be a valuable addition to the analysis. Unfortunately, these data were not systematically collected in our cohort and are therefore not available for inclusion. We have noted this as a limitation in the discussion.
Reviewer 3, comment 3:
The inclusion of one patient treated with anti-CD20 in the cohort is not informative within the context of this study. I suggest excluding this case from the study population and perhaps mentioning it in the discussion if deemed relevant.
Response 3: Thank you for this thoughtful suggestion. We agree that the patient in anti-CD20 therapy might influence group comparisons. Our inclusion criteria were patients with CIDP/MMN regardless of treatment type. To address the potential influence of this single case, we performed sensitivity analyses excluding the patient on CD-20-depleting treatment. The results were similar and are presented in the Results. We have clarified this and added a brief note in the Discussion to highlight that our main conclusions are robust to the exclusion of this case, while early IgG differences are sensitive to the exclusion of the patient in CD20-depleting treatment.
Reviewer 3, comment 4:
The criteria used to define an "adequate response" are only referenced from previous studies. I believe this is an important point that requires a more detailed explanation within the text.
Response 4: We agree that the rationale for our definition of an “adequate response” warrants further clarification. In our study, this term refers to the positive threshold of the assay for IgG against the wildtype SARS-CoV-2 RBD (≥225 AU/mL), as established in early validation studies. To avoid possible confusion, we have replaced “adequate response” with “positive response” throughout the manuscript. The same antigen (wildtype RBD) and assay were used consistently from 2021 to 2023, so the cut-off remained unchanged across the study period. While viral variants have emerged during this time, our measurements reflect antibody levels against the wildtype RBD, and we have acknowledged this as a limitation in the discussion. The selected cut-off, with a sensitivity of 94.3% and specificity of 99.5%, was kept unchanged as it reliably indicates the presence of IgG after vaccination and/or infection.
Reviewer 3, comment 5:
We can observe in the supplementary table that only a small number of participants were vaccinated with AstraZeneca ChAdOx1. How are these cases distributed among the three groups (MMN, CIDP, and controls)?
Response 5: All 21 participants who received a first dose of AstraZeneca ChAdOx1 were in the control group (21/304 controls), with none in the CIDP or MMN groups. Given the small proportion combined with the mRNA vaccines, we do not expect this to impact our results.
Reviewer 3, comment 6:
Figures 1 and 2 should include indicators of statistical significance, and their legends should be modified accordingly.
Response 6: Thank you for the suggestion. We have added indicators of statistical significance to Figures 1 and 2. Statistical significance of group differences at each visit is represented with stars: * = p < 0.05, ** = p < 0.01, *** = p <0.001. We have revised the corresponding figure legends accordingly.
Round 2
Reviewer 2 Report
Comments and Suggestions for Authors
The authors have carefully revised the manuscript and addressed the concerns raised in the previous review. Model assumptions and group imbalance have been clarified with appropriate diagnostics, and the statistical approach is now better justified. The explanation that linear mixed-effects models inherently account for repeated measures is acceptable. The figure legend for Figure 1 has been clarified, and terminology has been improved by replacing “adequate” with “positive response,” with appropriate acknowledgment of assay limitations and variant evolution. References have been corrected, and the discussion and conclusion have been revised to avoid overstated claims. Overall, the revisions substantially strengthen the manuscript.
As a minor point, I recommend a careful editorial review to correct remaining typographical issues (e.g., double spaces, double periods) and to ensure consistency of terminology throughout. Also, please separate figure legend for Figure 1 from main manuscript. Currently, they are merged.
Author Response
Reviewer 2, comment:
The authors have carefully revised the manuscript and addressed the concerns raised in the previous review. Model assumptions and group imbalance have been clarified with appropriate diagnostics, and the statistical approach is now better justified. The explanation that linear mixed-effects models inherently account for repeated measures is acceptable. The figure legend for Figure 1 has been clarified, and terminology has been improved by replacing “adequate” with “positive response,” with appropriate acknowledgment of assay limitations and variant evolution. References have been corrected, and the discussion and conclusion have been revised to avoid overstated claims. Overall, the revisions substantially strengthen the manuscript.
As a minor point, I recommend a careful editorial review to correct remaining typographical issues (e.g., double spaces, double periods) and to ensure consistency of terminology throughout. Also, please separate figure legend for Figure 1 from main manuscript. Currently, they are merged.
Author response: We sincerely thank the reviewer for the positive evaluation of our revisions and for the constructive suggestion. We have carefully proofread the manuscript to correct typographical errors (e.g., double spaces, double periods) and to ensure consistency of terminology throughout. In addition, we have separated the figure legend for Figure 1 from the main manuscript as requested.
Reviewer 3 Report
Comments and Suggestions for Authors
I think the manuscript has been sufficiently improved, I would only suggest to include in the text a comment answering to my fifth consideration on the type of vaccination into the study cohort.
Author Response
Reviewer 3
Reviewer 3, comment:
I think the manuscript has been sufficiently improved, I would only suggest to include in the text a comment answering to my fifth consideration on the type of vaccination into the study cohort.
Author response: We thank the reviewer for this helpful suggestion. In line with the comment, we have now added a statement in the Discussion (lines 402-405) clarifying that all 21 participants who received a first dose of AstraZeneca ChAdOx1 were in the control group, with none in the CIDP or MMN groups, and that this small proportion is unlikely to have impacted the study results.